# Evaluating competition for forage plants between honey bees and wild bees in Denmark

Claus Rasmussen[1]*, Yoko L. Dupont[2], Henning Bang Madsen[3], Petr Bogusch[4], Dave Goulson[5], Lina Herbertsson[6], Kate Pereira Maia[7], Anders Nielsen[8], Jens M. Olesen[9], Simon G. Potts[10], Stuart P. M. Roberts[11], Markus Arne Kjær Sydenham[12], Per Kryger[13]

1 Department of Agroecology, Aarhus University, Tjele, Denmark, 2 Department of Bioscience, Aarhus University, Kalø, Denmark, 3 Department of Biology, University of Copenhagen, Copenhagen, Denmark, 4 Faculty of Science, University of Hradec Králové, Hradec Králové, Czech Republic, 5 School of Life Sciences, University of Sussex, Brighton, United Kingdom, 6 Centre for Environmental and Climate Research, Lund University, Lund, Sweden, 7 Institute of Biosciences, University of Sao Paulo, Sao Paulo, Brazil, 8 Norwegian Institute of Bioeconomy Research (NIBIO), Ås, Norway and Centre for Ecological and Evolutionary Synthesis (CEES), Department of Biosciences, University of Oslo, Oslo, Norway, 9 Department of Biology, Aarhus University, Aarhus, Denmark, 10 Centre for Agri-Environmental Research, School of Agriculture, Policy and Development, University of Reading, Reading, United Kingdom, 11 Centre for Agri-Environmental Research, School of Agriculture, Policy and Development, University of Reading, Reading, United Kingdom, 12 Norwegian Institute for Nature Research (NINA), Oslo, Norway, 13 Department of Agroecology, Entomology and Plant Pathology, Aarhus University, Slagelse, Denmark

* claus.rasmussen@agro.au.dk

**Data Availability Statement:** All relevant data are within the manuscript and its Supporting Information files.

## Abstract

A recurrent concern in nature conservation is the potential competition for forage plants between wild bees and managed honey bees. Specifically, that the highly sophisticated system of recruitment and large perennial colonies of honey bees quickly exhaust forage resources leading to the local extirpation of wild bees. However, different species of bees show different preferences for forage plants. We here summarize known forage plants for honey bees and wild bee species at national scale in Denmark. Our focus is on floral resources shared by honey bees and wild bees, with an emphasis on both threatened wild bee species and foraging specialist species. Across all 292 known bee species from Denmark, a total of 410 plant genera were recorded as forage plants. These included 294 plant genera visited by honey bees and 292 plant genera visited by different species of wild bees. Honey bees and wild bees share 176 plant genera in Denmark. Comparing the pairwise niche overlap for individual bee species, no significant relationship was found between their overlap and forage specialization or conservation status. Network analysis of the bee-plant interactions placed honey bees aside from most other bee species, specifically the module containing the honey bee had fewer links to any other modules, while the remaining modules were more highly inter-connected. Despite the lack of predictive relationship from the pairwise niche overlap, data for individual species could be summarized. Consequently, we have identified a set of operational parameters that, based on a high foraging overlap (>70%) and unfavorable conservation status (Vulnerable+Endangered+Critically Endangered), can guide

**Funding:** PK received funding from Ministry of Environment and Food of Denmark (33010-NIFM-19-740). Funders played no role in the study design, data collection and analysis, decision to publish, or preparation of the manuscript.

**Competing interests:** The authors have declared that no competing interests exist.

both conservation actions and land management decisions in proximity to known or suspected populations of these species.

## Introduction

Many insects, and perhaps wild pollinators in particular, have recently declined at both local and regional scales in north-western Europe and North America [1–5]. As pollination is a vital ecosystem service, such findings are worrying [6]. The scientific evidence for decline is diverse, from local to global in scale and spans population monitoring [7–11], pollen-load analysis, and genetic information [12]. Conclusions are dire and mostly support the overall findings by Powney *et al.* [13], showing evidence of a historical decline of pollinators. Faced with this concern, new land use-management policies supporting wild pollinating insects are being implemented, but one of the recurrent concerns has been the potential competition for flower resources between wild bees and managed honey bees [e.g., 14–16]. Competition between species results from exploitation of the same limited resource and is common in plant and animal communities. In natural populations, such inter-specific competition leads to niche differentiation over evolutionary time, sometimes observed as character displacement, which acts to minimize competitive overlap between the species [e.g., 17]. In consequence, inter-specific interactions such as competition, together with predation, herbivory, mutualism, pathogenic interaction, and parasitism, all contribute to shaping distribution and abundance of organisms [18] ultimately structuring ecological communities.

Flower resources are critical to bee survival. Whether bees provide for their own nests or act as kleptoparasites that usurp the nests of other bees, the offspring almost exclusively and invariably develop upon stored pollen, an essential source of proteins and lipids, in addition to carbohydrates in the form of nectar [19, 20]. Adult bees mostly forage on nectar, to a varying extent on pollen, and in rare occasions on alternative resources like floral oils [19]. In plant-insect pollination networks, either pollinators or plants can be the limiting resource. For plants, a shortage of suitable pollinators leads to insufficient or a reduced quality of pollination that can ultimately affect reproduction [e.g., 21, 22]. For flower visiting insects, competition for food can lead to decline or extirpation of a species, through interference competition by aggressive foraging [23–25], or more often, by exploitative competition, following more efficient foraging by one or more pollinator species [26–28]. Because small populations, due to random demographic processes, are more likely to go locally extinct [29], flower visiting insects that are limited by e.g. nesting resources, in addition to floral resources, may be particularly vulnerable to competitive exclusion from species whose populations are only limited by floral resources [30].

Species of honey bees, *Apis* spp., are naturally distributed in the Palearctic region north from southern Norway and the Pacific maritime provinces of Russia and south through the entire African and Oriental regions [31]. In contrast to all other bees in Europe, honey bees have both a highly sophisticated system of recruitment and large perennial colonies where they store honey reserves for periods with reduced nectar supply. These reserves are the main reason for a long shared sweet history between honey bees and humans [32, 33]. Today nearly all European honey bees are managed [34, 35] and they are considered as important pollinators in both natural and agricultural systems [36]. While species of honey bees, *Apis* spp., very rarely behave aggressively towards other bees [25, 37], their recruitment system, large colony size, and the often high density of managed hives have led to concerns about possible resource

competition with other pollinators, in particular with native bees in areas where the western honey bee, *Apis mellifera*, has been introduced [38–45]. Recent studies have addressed the role of *A. mellifera* competition with other bees, both outside [e.g., 46] and within [e.g., 47] their native range. Although scenarios are not comparable across ecosystems, the conclusion is that high-density beekeeping generally triggers interspecific foraging competition, possibly depressing species richness [48] and local populations of wild bees [30], although landscape heterogeneity may modify the effect [49].

Wild bees not only depend on floral resources but also on availability and persistence of suitable sites for nesting and hibernation [50–53]. Competition is therefore affected by several, and very different resources, which inevitable vary at both spatial and temporal scales [30, 49, 54]. Consequently assessing the impact of honey bee competition on the population dynamics of wild bees can be highly challenging [see perspective in 55]. One route to improve our comprehension of this competitive interaction is to focus on the more detailed mechanisms of its drivers, in particular, the diversity of plant resources being utilized [56]. Here, we do this at a national scale by studying the interactions between managed honey bees and wild bee species in Denmark. Our focus is on floral resources shared by honey bees and wild bees, with an emphasis on both threatened bee species and foraging specialist species, as we expect that small populations (*i.e.*, typical of threatened bees) and narrow niche widths (*i.e.*, typical of pollen specialists) are the most sensitive to interspecific competition, including that from honey bees. Identifying these species of concern from competition is an important step in supporting wild pollinating insects.

## Material and methods

To evaluate the potential for food competition, we first collated all available data about forage plants for 292 bee species in Denmark and their national Red List conservation status. We then estimated pairwise foraging niche overlap and analyzed the bee-plant interaction network to assess the extent to which bee species were likely to be subjects to competition from honey bees.

We extracted the current list of the 292 Danish bee species, including their conservation status and taxonomy from Madsen [57]. See IUCN [58] for further detail on how these categories are defined and classified, as well as Madsen [57] for extensive annotations about the status of individual species in Denmark. 'Threatened species' includes those that are in the categories Critically Endangered (CR), Endangered (EN), and Vulnerable (VU), while 'Non-threatened' includes those that are Least Concern (LC) and Near Threatened (NT). Additional categories used are 'Data Deficient' (DD) and 'Regionally Extinct' (RE). Honey bees in Denmark are 'NA' (Not Applicable), *i.e.*, considered not eligible for a national assessment because of extensive management in the country [57].

While *A. mellifera* is native to Denmark, we do not include it here when referring to wild bees. The honey bee is the only bee species widely managed across the country [57]. Honey bees in Denmark are mostly a hybrid of several introduced subspecies, whereas the native dark Nordic *A. m. mellifera* subspecies is today only kept on the eastern part of the island of Læsø, with few other colonies maintained around the country [59]. Commercially reared bumble bee colonies of *Bombus terrestris* and mason bees (*Osmia bicornis* and *O. cornuta*) are also available for field and greenhouse pollination in Denmark. The extent of introgression from these and native populations is unknown but, by far, the majority of *B. terrestris* are wild [60].

We gathered information on all known forage plants of the bees in Denmark. Forage plants were here defined as plants visited for pollen, nectar, or both. For wild bees, we extracted information on all forage plants from neighboring German species-level records [61–64], in

addition, to a few additional records from the United Kingdom [65]. This set of references emphasized pollen sources for nest-building species, as well as preferred nectar sources for all species, including the kleptoparasitic species. For the honey bee, we extracted data on Danish forage plants from literature records and the unpublished "C.S.I. Pollen" study by the Danish Beekeepers' association. Honey bee literature records included all pollen and nectar food sources listed by Boelt [66] and Danmarks Biavlerforening [67], which are updated versions of the Danish honey bee plants list originally compiled by Christensen [68]. These references cover all known plants visited by honey bees as confirmed by observations and/or samples by beekeepers in Denmark. We also included the additional food sources listed by Kryger *et al.* [69], which emphasize crops with a dependence on insect pollination. The list of plants used for nectar and pollen collection by honey bees is extensive compared to the other species, reflecting their extensive active season, their generalized collecting of pollen and nectar, but also the relatively intensive sampling effort of honey bee resource utilization compared to that of wild bees. In addition to honey bee literature records, we also included the C.S.I. study, in which pollen from honey bee colonies were collected from 24 locations across Denmark, using hive mounted traps every third week from April to September in 2014 and 2015 [for detailed methods see 70]. Pollen in these samples were identified to species, genus, or family level. In a few cases, identification was only to a genus or a family aggregate due to low morphology-based pollen differentiation. Trace amounts of pollen (<5 grains/500 grains) were excluded from the list. To avoid synonyms and nomenclatorial differences across sources, we updated all plant names across sources to reflect the taxonomy used in the Danish national database [71]. Species not listed in this database, mostly ornamental and/or non-native plants, were cross-checked with The Plant List [72]. Family level was based on The Plant List [72].

Pollen specialization to a single plant species is very rare and floral specialization is often for all or several species within a plant genus, multiple genera or related families [19, 73]. Thus, in our study, we have excluded species-level plant information and only compared forage plants at the genus level for each of the Danish bee species. Such a conservative approach implies that the record of a forage plant species means that any species, within that plant genus and irrespectively of the number of species in the genus, will serve as forage plants for a given bee species. As an example, *Vicia* (Leguminosae) has bee records across all sources from *V. cracca*, *V. faba*, *V. hirsuta*, *V. nigra*, *V. onobrychioides*, *V. sativa*, *V. sepium*, *V. tenuifolia*, *V. villosa*, and unidentified species of *Vicia*. These are here pooled together as the genus *Vicia*. As noted by [74], those species that consistently collect pollen only from the same single species of floral host are considered "a curiosity" with little biological meaning. In addition to conservation status and number of forage plants, we also recorded pollen generalization level of each bee species as polylecty, if the bee was without any preferred plant affinities or oligolecty, if the pollen affinities were limited to a narrow range of plant genera or families [sensu 74]. For further details, see appendix 1. Kleptoparasitic species, that usurp nests of other bees, do not actively collect pollen. They are therefore not recorded as being either polylectic or oligolectic, but classified as kleptoparasitic. Forage plants for kleptoparasitic bees include only those which adult bees have been documented to visit for nectar, and not the plants on which their hosts depend and their larvae forage on. Kleptoparasitic species may still indirectly share the specialization of pollen with their hosts, but relationships are often equivocal. For instance, a single kleptoparasitic species can usurp the nests of both polylectic and oligolectic species, *e.g.* *Sphecodes rubicundus* in nests of *Andrena flavipes* (polylectic) and *A. labialis* (oligolectic), and *Stelis punctulatissima* in nests of *Osmia aurulenta* (polylectic) and *O. leaiana* (oligolectic). Some kleptoparasites have multiple hosts that are all oligolectic, such as *Nomada flavopicta*, that can usurp nests of the four Danish species of the oligolectic *Melitta*. These four species are

each specialized on different, unrelated plant genera. Status as kleptoparasitic is based on Michener [31] and Scheuchl and Willner [63].

## Pairwise niche overlap

From the recorded interactions between bee species and forage plant genera, we summarized and calculated the following measures for each bee species: Total number of known forage plants; Number of forage plants only visited by each individual wild bee species, but not by honey bees (no overlap); Number of forage plants visited by both the individual wild bee species and honey bee (overlap); and MacArthur and Levins [75]'s asymmetrical measure for pairwise niche overlap, specifically the estimate of the extent to which honey bees overlap with wild bee species ($M_{kj}$) [76]:

$$\widehat{M}_{kj} = \sum_i^n \hat{p}_{ik}\hat{p}_{ij} / \sum \hat{p}_{ik}^2$$

$M_{kj}$ is pairwise niche overlap of honey bee species $k$ on bee species $j$, $p_{ik}$ is the proportion that resource $i$ comprises of the total resources used by honey bee species $k$, $p_{ij}$ is the proportion that resource $i$ comprises of the total resources used by the wild bee species $j$, and $n$ is the total number of resources (forage plants). As the relative proportion of the forage plants in the diet is unknown for most of the bee species in our data, besides being both temporally and spatially variable, only presence-absence data (whether or not the plants served as food) has been used. Other obvious measures, *e.g.* Pianka [77]'s symmetrical measure for pairwise niche overlap, are dependent on quantitative data and are therefore not used here.

## Overlap of modules in the bee-plant network

As all plants and bees in our dataset are connected through pollination interactions, the system can be described as a bipartite mutualistic network [78]. Thus, a pollination network was assembled as an incidence matrix of interacting plant genera (columns) and bee species (rows). A cell entry $x_{ji}$ had a value of one if bee species $j$ (either a wild or honey bee) visits flowers of plant genus $i$, *i.e.*, if an interaction is present, but zero if an interaction is absent. Such networks often display sub-structural patterns, when some species/genera interact strongly with each other, but less so with the remaining species, thereby creating densely connected sub-groups (modules) of bee species and plant genera (collectively all taxa are termed nodes) within a less connected network [79]. Modularity is a measure of the extent to which the interactions are concentrated in modules rather than among different modules, compared to null model networks [80]. In the context of bee diet, bee species in the same module are expected to share more forage plants than with bee species in other modules. Hence, competition based on shared plant genera, is expected to be higher among members within modules, rather than across modules. Therefore, bee species placed in the same module as honey bees may be more prone to competition due to diet overlap, than bee species in other modules. In this analysis, we considered only the plant genera visited by wild bees, *i.e.* the plant genera recorded only for honey bees were excluded, as these plants do not contribute to the diet of wild bee species nor to the pairwise niche overlap measure. Modules in the plant-bee network were identified by optimizing the Barber modularity metric $Q_b$ using an optimization algorithm developed for bipartite networks [81]. The algorithm uses an iterative heuristic search for maximizing $Q_b$, in order to identify an optimal partition of the network in modules [for further information about the method, see 79, 81, 82]. The software *MODULAR* [82] was used to calculate $Q_b$, and to assess its significance against 100 randomizations of the network. The null model

implemented in the program generates random matrices conserving the row and column totals of the empirical network [null model 2 from 83].

Due to the heuristic nature of the analysis, network partitioning varies among different runs using the same input matrix. Hence, we repeated the modularity analysis 100 times, in order to assess variability in modularity $M$, number of modules $N_M$, and module composition. Specifically, we counted the number of runs each node (bee species or plant genus) was placed in the same module as the honey bee. As nodes within modules interact more strongly with each other than with nodes in other modules, bee species repeatedly placed in the same module as honey bees are potentially most strongly affected by competition through shared diets.

All summary statistics were done in Excel, JMP® 14.0.0, and RStudio, including package 'Bipartite' [84–88].

## Results

Across all 292 known bee species from Denmark, a total of 410 plant genera were recorded as forage plants. These included 294 plant genera visited by honey bees and 292 plant genera visited by wild bees. Ten bee species did not have any recorded forage plants from Germany in the examined literature: *Andrena albofasciata*, *A. morawitzi*, *Bombus quadricolor*, *B. veteranus*, *Hylaeus gracilicornis*, *H. pfankuchi*, *Lasioglossum sexmaculatum*, *L. sexnotatulum*, *Nomada moeschleri*, and *Sphecodes rufiventris*. Of these two, *B. quadricolor* and *H. pfankuchi*, were considered regionally extinct in Denmark. 118 plant genera were only known to be visited by honey bees, and not by wild bees, and 116 plant genera were only known to be visited by wild bees. This leaves 176 plant genera, where a foraging overlap occurred between honey bees and wild bee species. The bees included 148 polylectic, 65 oligolectic, and 79 kleptoparasitic species. Out of these 292 bee species, 56 species were classified threatened (VU+EN+CR) in the Danish Red List, 26 are classified as NT, 137 as LC (latter two 'not threatened' category), 48 as NA, 6 as DD (latter two 'other' category), and 19 RE ('regionally extinct' category) (Appendix 1).

### Honey bees

Honey bees (*A. mellifera*) forage on a diverse range of plants (Appendix 3), in particular for nectar [67], but much of the season, workers also collect pollen from a narrow plant selection (C.S.I. unpublished data). Half of all samples in 2014 and 2015 contained some pollen from 21 plant genera, in particular the often mass-flowering genera *Trifolium* (6.1% of the samples), *Taraxacum* (4.7%), and *Pyrus* (3.4%). This pollen occurrence was number of times a plant taxon has been recorded (presence/absence) in any sample across time and space. No such quantitative data exists for nectar sampling. If honey bee-plant genera were assessed by relative importance, *i.e.*, taking into account the proportion of specific pollen grains in each sample, then honey bees appeared even more specialized on a narrow selection of plant genera, dominated by *Trifolium* (19.1%), along with *Salix* (8.8%), *Pyrus* (7.3%), *Brassica* (6.3%), *Taraxacum* (5.1%), and *Acer* (3.8%). All other forage plants were much less common in the samples, although we have not been able to take into account the variable size of pollen grains and the potential over-representation of certain species. If we compared all pollen samples across time and space, 29 plant genera dominated in the samples (having >50% of the grains). Some of those 29 plants important to the honey bees were locally and temporarily restricted, including summer flowering outliers *Rubus* (only found in four samples and only one >50%) and *Artemisia* (11 samples and only one >50%). We also observed that honey bees used a larger diversity of forage plants later in the season, whereas in the beginning of the season they were more often using few plant genera (Fig 1).

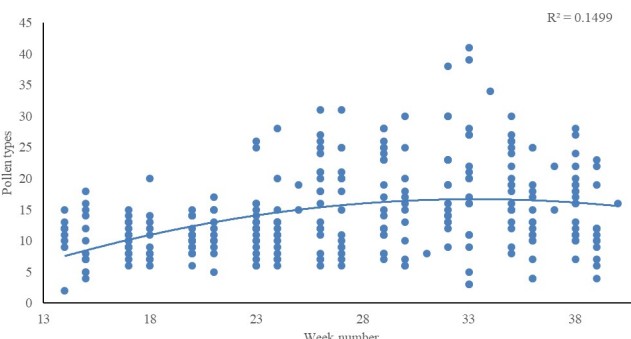

**Fig 1. Number of different pollen types present in multiple pollen samples from across Denmark as a function of week number 14 to 40 (C.S.I. data merged for 2014 and 2015).** A polynomial trend line ($R^2 = 0.1499$) for the data is added.

## Wild bees

Early-flowering willows (*Salix* spp.) were important to 80 wild bee species, but even more so were later-flowering *Taraxacum*, *Cirsium*, and *Rubus* in late spring and summer with 107, 95, and 92 species of wild bees, respectively, foraging on these genera.

## Pairwise niche overlap

Frequency distribution of links per taxon was skewed, *i.e.*, a few plant genera and bee species had many interactions while most bees and plants had few interactions. Bees had an average of 12.4 ± 19.0 interactions, including the honey bee and 11.4 ± 8.8, excluding it. Plants had an average of 8.5 ± 15.0 interactions, including the honey bee and 11.0 ± 16.7, excluding it along with the 118 plants exclusive to honey bees. A high degree of overlap in the interactions was found, both in terms of plants that were shared as forage plants among different bee species, but also in terms of visiting bees that were shared among plant genera.

Resource overlap between honey bee and individual wild bee species ranged from 0% to 100%. No overlap was observed for the oligolectic *Melitta tricincta* foraging on *Odontites* from which honey bees were not reported. An overlap of 100% in as many as 61 species of wild bees included both bees with very few (down to one) known forage plants and others like *Andrena fulvida* with 18 known forage plants, all of which were also forage plants for honey bees (Appendix 1).

Of the 292 wild bee species in Denmark, 200 had more than a 70% overlap of their forage plants with honey bees (Fig 2). An overlap of 70% meant that of all of the forage plants utilized by a wild bee species, 70% of those plants had the potential to be shared with honey bees. Whether or not plants were shared with other wild bee species, *i.e.*, potential competition interactions among wild bee species were not addressed by pairwise overlap.

Total number of food plants per bee species, excluding the honey bee, differed highly significantly among polylectic, oligolectic, and kleptoparasitic species (Kruskal-Wallis: $H = 62.47$, $df = 2$, $p < 0.0001$) and genera (Kruskal-Wallis: $H = 53.27$, $df = 31$, $p < 0.01$). Red List categories also differed in total number of food plants (Kruskal-Wallis: $H = 30.15$, $df = 7$, $p < 0.001$), with fewest food plants among RE (average 5,42) and most food plants among LC (13.21) and DD (12.17). Total number of food plants and pairwise niche overlap were highly correlated ($r_s = -0.17$, $p < 0.01$).

When comparing pairwise niche overlap ($M_{kj}$) with foraging specialization (lecty) and Red List status, we found that 11 threatened (CR, EN and VU) bee species shared at least 90% of their forage plants with the honey bee, and 30 threatened species shared at least 70% of their forage

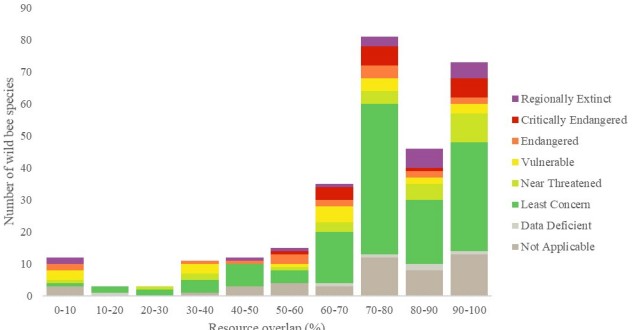

**Fig 2. Histogram with the number of wild bee species (y-axis) that has foraging overlap in percent with honey bees in intervals of 10 (x-axis).** The honey bee is excluded and intervals are $90 < i \leq 100$, etc. except $0 \leq i \leq 10$. 'Threatened species' includes those that are in the IUCN categories Critically Endangered (CR), Endangered (EN), and Vulnerable (VU), while 'Non-threatened' includes those that are Least Concern (LC) and Near Threatened (NT). Additional categories used are 'Data Deficient' (DD) and 'Regionally Extinct' (RE). Honey bees in Denmark are 'NA' (Not Applicable), i.e., considered not eligible for a national assessment because of extensive management in the country.

plants with the honey bee (Table 1). However, level of pairwise niche overlap, excluding the honey bee, did not differ significantly among polylectic, oligolectic, and kleptoparasitic species (Kruskal-Wallis: $H = 0.16$, $df = 2$, $p = 0.92$) (Fig 3), families (Kruskal-Wallis: $H = 7.51$, $df = 5$, $p = 0.19$) or genera (Kruskal-Wallis: $H = 36.09$, $df = 31$, $p = 0.24$), nor did Danish Red List categories differ in level of pairwise niche overlap (Kruskal-Wallis: $H = 11.99$, $df = 7$, $p = 0.10$).

## Overlap of modules in the bee-plant networks

In the network, excluding 118 plant genera visited exclusively by the honey bee, $I = 3,378$ bee-plant interactions were listed between $B = 282$ bee species and $p = 292$ plant genera visited by wild bee species. Hence, connectance $C = 100\ I/(BP) = 4.10\%$. Plant genera interacted with an average ± SD of 11.57 ± 16.82 (range 1–109) bee species; the most generalized plant genus being *Taraxacum*. Bee species interacted with an average of 11.98 ± 13.16 plant genera (range 1–176), with *Apis mellifera* being the most generalized species, far exceeding the second most generalized bee species, *Andrena flavipes* (58 interactions). The network was significantly modular ($p < 0.0001$), when compared to null networks. The level of modularity was 0.373 ± 0.004 ($N = 100$ runs), and the network was partitioned into an average of 5.16 modules (4 modules in 3 runs, 5 modules in 78 runs, 6 modules in 19 runs) (Fig 5).

A total of 35 plant genera shared their module with the honey bee in all runs of the modularity analysis, while 149 plant genera never did so (Fig 4A). A total of 219 bee species were

**Table 1. Number of total extant species (Danish Red List categories NA+DD+LC+NT+VU+EN+CR), not threatened species (LC+NT) and threatened species (VU+EN+CR) that share 50%, 70% or 90% of their food sources with honey bees.**

|  | Extant | | | Not Threatened | | | Threatened | | |
|---|---|---|---|---|---|---|---|---|---|
|  | >50% | >70% | >90% | >50% | >70% | >90% | >50% | >70% | >90% |
| All | 241 | 190 | 70 | 146 | 122 | 44 | 47 | 31 | 11 |
| Polylectic | 130 | 100 | 27 | 83 | 68 | 20 | 23 | 12 | 2 |
| Oligolectic | 46 | 39 | 25 | 21 | 20 | 12 | 8 | 6 | 5 |
| Kleptoparasitic | 65 | 51 | 18 | 42 | 34 | 12 | 16 | 13 | 4 |

Categories are for all bees, but also divided for polylectic, oligolectic and kleptoparasitic species. The latter does not collect pollen, but usurp the nest and consume the pollen of its host. All 19 regionally extinct species and the honey bee have been excluded from the present table.

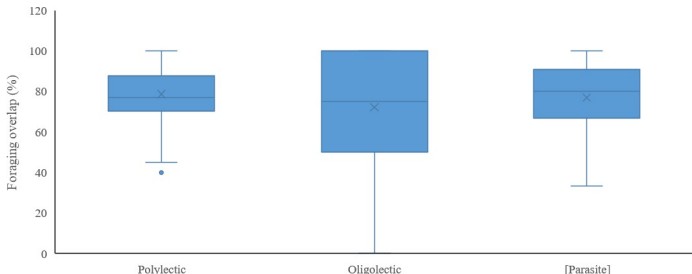

**Fig 3. Box-plot of different functional groups of wild bees and their foraging overlap with honey bees.** The honey bee and wild bees with no recorded forage plants are excluded.

never included in the same module as the honey bee, while no bee species were always included (Fig 4B). However, three bee species (*Lasioglossum sexnotatum*, *Macropis europaea*, and *Melitta nigricans*) were often placed in the same module as the honey bee (in >95 runs of 100). Another seven bee species (*Hylaeus pectoralis*, *H. punctulatissimus*, *Lasioglossum nitidulum*, *L. morio*, *Andrena nigrospina*, *Sphecodes marginatus*, and *Bombus terrestris*) sometimes shared the module with *Apis mellifera* (>50 but < 95 runs of 100). A total of 18 bee species were rarely (>10 but < 50 runs of 100), and nine bee species were very rarely (<10 runs of 100) placed in the honey bee module.

Level of module overlap did not differ among oligolectic, polylectic and kleptoparasitic species (Kruskal Wallis $H = 4.36$, $df = 2$, $p = 0.11$), nor among Red List categories (Kruskal Wallis $H = 9.84$, $df = 7$, $p = 0.20$). A significant difference in module overlap was found between different bee families (Kruskal Wallis $H = 20.88$, $df = 5$, $p < 0.001$), and genera (Kruskal Wallis $H = 81.63$, $df = 31$, $p < 0.001$). A pairwise comparison using Tukey-Kramer HSD revealed that Melittidae had a significantly higher module overlap with honey bees than the remaining bee families. In particular, the genus *Macropis* had a significantly higher module overlap compared to 19 other bee genera (*Hylaeus*, *Lasioglossum*, *Andrena*, *Megachile*, *Bombus*, *Halictus*, *Nomada*, *Colletes*, *Osmia*, *Dasypoda*, *Anthophora*, *Stelis*, *Coelioxys*, *Sphecodes*, *Melecta*, *Anthidium*, *Hoplitis*, *Epeolus* and *Panurgus*), while *Melitta* had a significantly higher module overlap only with *Andrena* and *Nomada*.

Pairwise niche overlap and module overlap were not significantly correlated ($r_s = 0.00$, $p = 0.99$), nor was total number of food plants and module overlap ($r_s = 0.01$, $p = 0.85$).

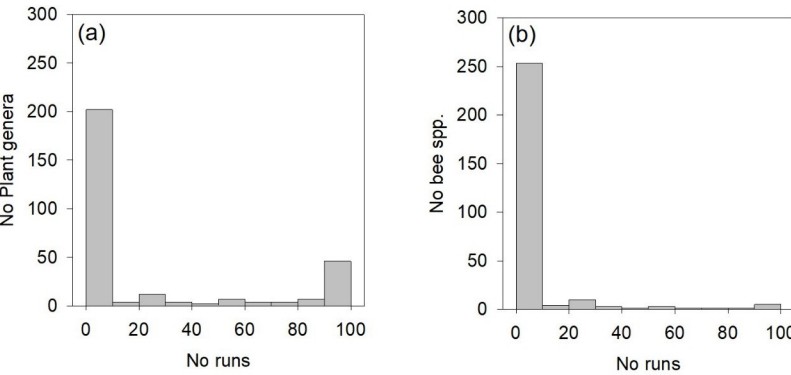

**Fig 4.** Number of runs in which (a) plant genera or (b) bee species is placed in the same module as honey bees.

## Discussion

Honey bees are both abundant and widespread in Denmark where they are native, but today mainly managed for honey production and crop pollination. During the season, a single colony may collect 650,000 pollen loads, equivalent to 110,000 solitary bee progeny [89]. Such high figures have prompted the necessity to assess the niche overlap between honey bee species and wild bee species and may ultimately require that managed, not feral, honey bees in apiaries are regulated in numbers or distribution near populations of threatened bee species. Here, we focused upon relationships between wild bees and honey bees. In order to do so, we mapped interactions between the Danish wild bee species and plant genera, addressing diet overlap with honey bees. We identified a set of bee species that are assessed as threatened for the Danish Red List and have a potentially high (>70%) or very high overlap (>90%) of forage plants with honey bees. This overlap was unrelated to diet width of the wild bees, *i.e.*, whether they were oligolectic, polylectic, or kleptoparasitic. However, oligolectic species may have fewer alternative forage plants to turn towards, if honey bee competition intensifies [but see 90] and consequently, those oligolectic species that share most of their forage plants with honey bees, could be particularly sensitive to competition. This we could not address directly with our dataset. Within the oligolectic group of species, we identified, for example, five threatened species that shared >90% of their forage plant genera with honey bees (*Andrena lathyri*, *Dasypoda suripes*, *Dufourea halictula*, *Dufourea inermis*, and *Hoplitis anthocopoides*). These five species are all very specialized, both regarding the pollen source and habitat type, and their long-term survival may benefit from a removal of nearby honey bee hives. In addition to this, any negative effects of a presence of honey bees could be mitigated by boosting the abundance of common forage plants. In general, these high overlaps and oligolectic bee species should be taken into consideration in local conservation plans to promote viable populations. It is important to stress that according to the Danish Red List data, none of the species entries are considered subject to food competition [57], but it is their current status that makes them susceptible to additional threats including resource limitations. Pairwise foraging overlap with honey bees is not significantly correlated with status on the Danish red list, thus, there is no reason to suggest that competition *per se* from honey bees has resulted in the species being added to the Red List.

While significant correlations exist between total number of food plants and certain parameters, including lecty and Danish Red List status, it is important to notice that this is not necessarily related to foraging overlap, as polylectic species for example on average simply have a higher total number of forage plants.

The network analysis of the Danish bee-plant network suggested that the honey bee is a subject on its own. The module containing the honey bee is more isolated from other modules, *i.e.*, with fewer links to other modules, while the remaining modules are more highly inter-connected (Fig 5A). Only a few wild bees were consistently placed in the module with honey bees, and at least some of these are somewhat aberrant foragers, including oil bees (*Macropis europaea*) common in humid areas with *Lysimachia*, and *Hylaeus pectoralis* (90 of 100 times) common in dense stands of reed (*Phragmites australis*) where they establish their nests. Neither habitats appear to be particularly attractive to honey bees, nor for establishing apiaries, although they are sometimes found in close proximity [91, 92] and all had high pairwise niche overlap with honey bees (77.8–100%). Potentially these species are placed in the honey bee module, simply because they do not fit any of the other wild bee modules. The modular structure of the bee-plant network and the isolated position of the honey bee module indicate there is an extensive overlap in diet among groups of species of wild bees in Denmark.

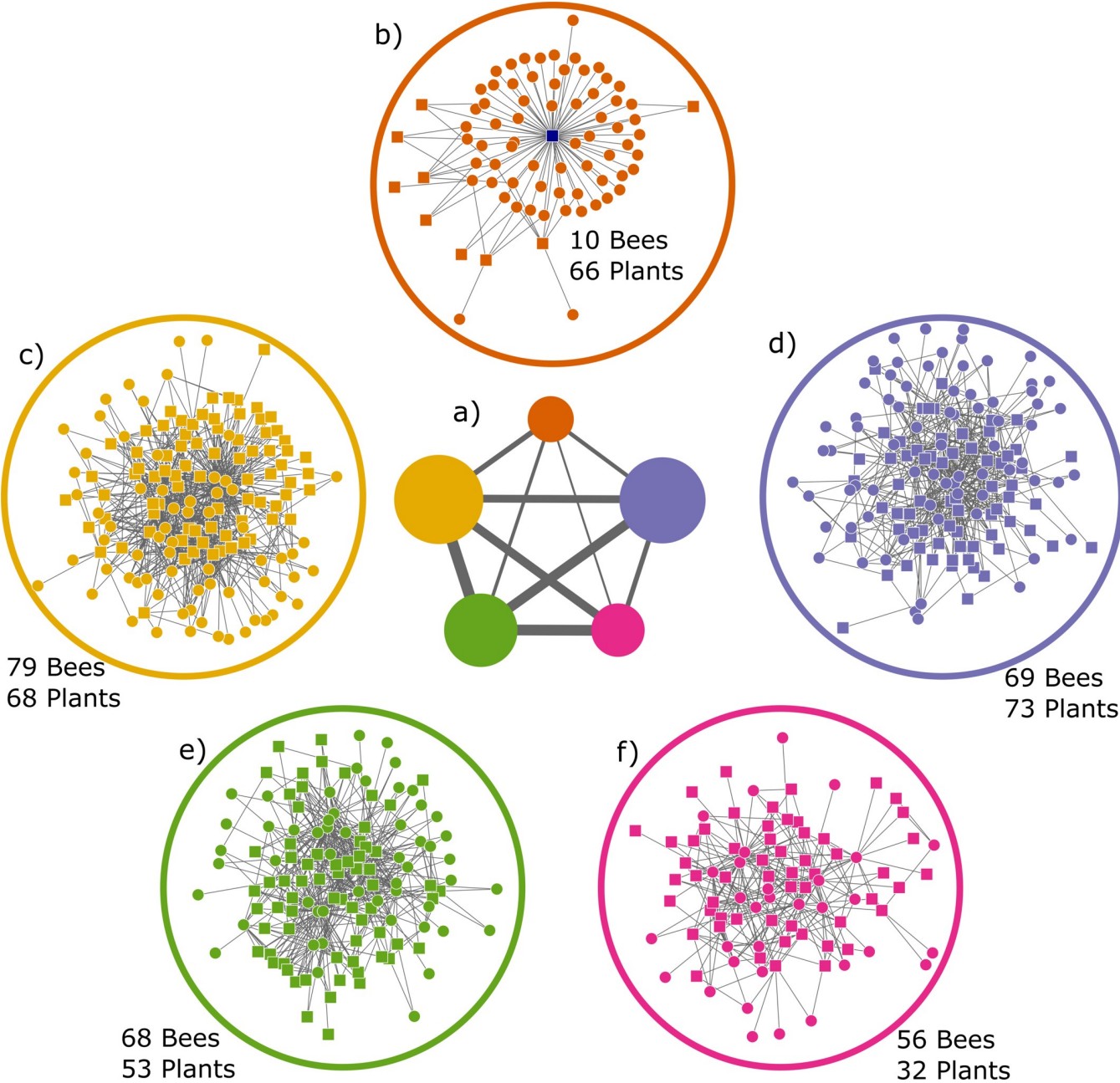

**Fig 5. Graph of the five modules.** a) Size of circles (modules) are proportional to number of both plant and bee species in each module, thickness of lines connecting modules are proportional to number of interactions between the modules. b-f) the five different modules identified from a, squares are bee species, circles are plant genera. Modules b to f are presented in the same position as they appear in a, with module b in top. Honey bee is the central square (blue) in the module b.

Using diet overlap as a measure of competition needs a cautionary note. Diet overlap may also be interpreted as a lack of interspecific competition, *i.e.*, species use the same resources, but do not interfere, especially if food resources are not the limiting factor. We assume, food resources in an intensive agricultural landscape often are limited, but we do not know [but see 93]. Competition effects on wild bees are therefore context dependent, and have been shown to depend on local density of honey bee hives [94–98]; distance from apiaries [30, 99]; resource

level in the landscape [49, 56; but see Lindström *et al.* 100], including for honey bees the availability of mass-flowering plants for nectar regardless if crops or wild flowers. Although honey bees can forage over distances up to several kilometers, they will forage within closer proximity of their hive if resources are available [101, 102]. Even spatiotemporal variability in availability of floral resources [e.g., 103], differences in response of bees to patch size and local flower density [104], and the phenology of bee species may influence the degree to which wild bees face competition from honey bees. Small-bodied bee species, even polylectic species, with limited foraging ranges may be more sensitive to competition than larger, wider-ranging species, if floral resources become exhausted at a local (< 250m) scale. Honey bees collect most of the pollen for the colony at the beginning of the season [105]; initial local competition may promote short-term floral resource switching, cascading competition to other forage plants [54]. The relative importance of nesting *versus* floral resources as a limiting factor for wild bees affects the sustainable honey bee density [56]; even for common and widespread species such as *Osmia bicornis*, the availability of suitable nesting sites may be the main limiting factor for population growth rather than forage plants [106].

The data we used has been collected in a non-systematic way. No comprehensive quantitative data on floral resource use exists for the entire wild bee fauna of Denmark making it also impossible to compare with the quantitative data for the honey bee pollen collection (the C.S.I. Pollen study). Honey bee nectar collection has also not been quantified by floral source. The data we use are therefore strictly presence/absence records. Obtaining quantitative data on flower use would help to inform further whether particular wild bee species are especially vulnerable to competition with honey bees, which would require more intensive study of their shared ecology. Quantitative estimates of the relative importance of different plant species for wild bees based on e.g. pollen samples or flower visitation frequencies would likely only reflect the realized and not the fundamental niche of the wild bee species, which could underestimate the foraging overlap between honey bees and wild bees. Definitive evidence of competition between bee species is notoriously hard to obtain, not least because bees are highly mobile and population dynamics of wild bees are difficult to measure [43, 44, 107]. While quantitative data would be useful, local conditions still vary over time and competition could go undetected if otherwise marginal food resources temporally become critical for the population. The data we used consists of recorded associations between wild bees and plants that have been accumulated over time. As such, and despite the limitations of binary data discussed above, we believe that our approach provides useful estimates of the fundamental niche of wild bees and of the potential niche overlap between wild and managed honey bees.

We also do not specifically consider interactions between honey bees and other bees mediated via spread of shared pathogens or parasites. There is a net flow of diseases from honey bee apiaries into nearby populations of bumble bees, with possible negative spill-over to other wild bees [108–112]. Transmission of pathogens between species is thought to occur largely through shared flower use [113, 114], thus, our analysis is likely to still identify wild bee species most at risk from being exposed to disease from honey bees [115], or the other direction, from wild bees to honey bees. Thus, our network visualization becomes a map of both potential food competition and pathogen cross-transmission. Our understanding of the host range of bee pathogens and the relative susceptibility of different bee species is still very poor [116, 117].

Only with regular surveys and more data on the threatened species in Denmark can it be ascertained whether competition plays a role and how regulation of hive density in natural areas affect the populations of wild bees. Perhaps the most promising experimental avenue to test for competition between honey bees and wild bees is replicated exclusion of honey bees, followed by monitoring of any response in wild bee populations.

## Conclusions

We do not aim to establish if competition between honey bees and wild bees is measurable in Denmark or at which densities of honey bees would be environmentally sustainable. Using a large dataset, we have identified a set of operational parameters that based on a high foraging overlap and unfavorable conservation status can guide both conservation action and land management plans. Thus, our study and its results provide the initial tools for decision-making. In particular, our individual species assessments identify species of concern as those with a suggested overlap above 70% and a Danish Red List status of threatened (VU+EN+CR). Until we fully understand the complex interactions between managed honey bees and threatened wild bee species, it would seem prudent to use the precautionary principle and avoid placing honey bee hives in places close to known or suspected populations of threatened bee species, during their active season, when there is a high niche overlap.

## Supporting information

**S1 Table. List of Danish species of bees, both wild and honey bees; foraging specialization [based on 63]; Danish Red List category [based on 57]; status as threatened (VU+EN+CR), not threatened (LC+NT), other (NA+DD) or Regionally Extinct (RE); total number of known Forage Plants (FP) for the bee species (0 when forage plants are unknown in the revised literature); number of the forage plants only visited by the wild bee species, but not by honey bees; overlap of forage plants that are visited both by wild bee species and by honey bees; MacArthur and Levins asymmetrical measure for niche overlap of honey bee species on wild bee species.** While the *Bombus lucorum*-complex in Denmark is near impossible to distinguish morphologically, in particular amongst the workers, they are here maintained separately because of certain known forage-differences [118]. *Andrena albofasciata* is considered a junior synonym of *A. ovatula* by [63, 64], but retained here in agreement with [57]. Lecty is not fully confirmed for *Andrena nanula* and *Lasioglossum sexnotatulum*. The former visits Compositae and Rosaceae in Denmark, although that might have been nectar sources, as it is suspected oligolectic on Apiaceae in neighboring countries [119]. The latter species is suspected polylectic based on forage plants in Finland [120]. Other species may be to strictly defined by Scheuchl and Willner [63], e.g. the polylectic *Andrena ovatula*, *Colletes cunicularis*, *Hoplitis leucomelana*, *Megachile circumcincta* and *M. lagopoda* which at least have specific pollen preferences [e.g., 121], and might be considered oligolectic.
(PDF)

**S2 Table. Full list of plant genera and their family.**
(XLSX)

**S3 Table. Full list of plant-bee relations.**
(XLSX)

## Acknowledgments

We are grateful to Danish Beekeepers' association for providing the dataset from the C.S.I. Pollen study.

## Author Contributions

**Conceptualization:** Claus Rasmussen, Yoko L. Dupont, Henning Bang Madsen, Per Kryger.

**Data curation:** Claus Rasmussen.

**Formal analysis:** Claus Rasmussen, Yoko L. Dupont, Kate Pereira Maia.

**Funding acquisition:** Per Kryger.

**Investigation:** Claus Rasmussen.

**Methodology:** Claus Rasmussen, Yoko L. Dupont, Henning Bang Madsen.

**Visualization:** Claus Rasmussen, Yoko L. Dupont, Kate Pereira Maia.

**Writing – original draft:** Claus Rasmussen, Yoko L. Dupont.

**Writing – review & editing:** Claus Rasmussen, Yoko L. Dupont, Henning Bang Madsen, Petr Bogusch, Dave Goulson, Lina Herbertsson, Kate Pereira Maia, Anders Nielsen, Jens M. Olesen, Simon G. Potts, Stuart P. M. Roberts, Markus Arne Kjær Sydenham, Per Kryger.

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
