## [Decision Letter · Decision Letter 0]

18 Mar 2021

PONE-D-21-02226

Evaluating competition for forage plants between honey bees and wild bees in Denmark

PLOS ONE

Dear Dr. Rasmussen,

Thank you for submitting your manuscript to PLOS ONE. After careful consideration, we feel that it has merit but does not fully meet PLOS ONE’s publication criteria as it currently stands. Therefore, we invite you to submit a revised version of the manuscript that addresses the points raised during the review process.

Please, I would recommend you follow referees' suggestion to improve your manuscript. There a minor but important issues that should be revised.

We look forward to receiving your revised manuscript.

Kind regards,

Fabio S. Nascimento

Academic Editor

PLOS ONE

Journal Requirements:

3. Please remove your figures from within your manuscript file, leaving only the individual TIFF/EPS image files, uploaded separately.  These will be automatically included in the reviewers’ PDF.

Reviewers' comments:

Reviewer's Responses to Questions

**Comments to the Author**

1. Is the manuscript technically sound, and do the data support the conclusions?

Reviewer #1: Yes

Reviewer #2: Yes

2. Has the statistical analysis been performed appropriately and rigorously? 

Reviewer #1: Yes

Reviewer #2: Yes

3. Have the authors made all data underlying the findings in their manuscript fully available?

Reviewer #1: Yes

Reviewer #2: Yes

4. Is the manuscript presented in an intelligible fashion and written in standard English?

Reviewer #1: Yes

Reviewer #2: Yes

5. Review Comments to the Author

Reviewer #1: I have reviewed the manuscript entitled "Evaluating competition for forage plants between honey bees and wild bees in Denmark". This is an interesting approach based on data available in the literature, which investigated floral resources shared by honeybees and wild bees, with an emphasis on both threatened wild bee species and foraging specialist species from Denmark. It is an original and very well written paper investigating the potential for food competition among wild bees and honeybees. The authors based their analysis on data available for 292 bee species. The conservation status of these species was also considered. I have two main concerns: i) the list of plants used by the 292 Danish bee species was restricted to plant genus, which would not be the most desirable for the best conclusion; ii) for wild bees, the authors extracted information on all forage plants from neighboring German species. Despite this, I still believe that the study is innovative and the results are relevant. Thus, I suggest accepting this paper after minor revision.

P.2 – L.70: to insufficient, or a reduced quality of, pollination – delete the comma changing to: to insufficient or a reduced quality of pollination

P. 2- L.75-76: Please give the proper credit to Henry and Rodet 2018 (Ref [46]). The idea in lines 75-76 was sustained by the study of these authors. Please, give them the credit.

P.6 – L.267: replace plans by plants

P.6 – L. 269-270: give emphasis to the negative association (rs) between the total number of food plants and pairwise niche overlap.

Figure 1 legend: polynomial trend line (R² = 0,1499) – replace comma by dot: 0.1499

Reviewer #2: This is a very interesting and elegant study evaluating competition between (managed) honey bees and wild bees over floral resources in Denmark. I applaud the aim of this study and using an impressive database, the authors analyse the data in a creative way, thereby providing useful guidelines for conservation efforts. The manuscript is beautifully written, clear and precise and the data properly analysed. I therefore highly recommend publication and I only have a few minor remarks.

L41: Consider changing: the word ‘network’ with ‘interactions’

L45: I would replace the abbreviations ‘VU+EN+CR’ by full names

L86: I think the references here should include ‘Santos et al., 2012’; or ‘Santos et al., 2012’ should replace reference ‘42’

L89: I suggest to rephrase to ‘depressing species richness (Angelella et al., 2021) and local populations’

L135: ‘honey’ should read ‘nectar’

L220-222: Consider to include these two sentences in Abstract. It is impressive that honey bees and wild bees share 176 plant genera in Denmark (almost 43% of known plant genera recorded as forage plants)

L266: Please, consider using “hereafter referred to as K-W’

L268, Appendix 1 (8th column): Replace comma by dot

L327: ‘nearby bee’ should read ‘nearby honey bee’

L340: ‘figure 5a’ should read ‘Figure 5a’

Fig. 1: Since there are 5 figures and 1 table, I would transfer this figure to ‘Supporting Information’ section

Figs. 1, 2, 3: Please, add axis titles

Fig. 2: I would replace the abbreviations by full names and add ‘Red list categories’ to the legend

Fig. 4: ‘plant genus’ should read ‘plant genera’ and ‘No Plant’ should read ‘No plant’

Fig. 5: I ‘redrew’ this figure (please find the attached file for your consideration)

References

Angelella G, McCullough C, O’Rourke M (2021). Honey bee hives decrease wild bee abundance, species richness, and fruit count on farms regardless of wildflower strips. Scientific reports 11, 3202.

Santos GMM, Aguiar CM, Genini J, Martins CF, Zanella FC, Mello MA (2012). Invasive Africanized honeybees change the structure of native pollination networks in Brazil. Biological Invasions 14, 2369-2378.

6. PLOS authors have the option to publish the peer review history of their article (what does this mean?). If published, this will include your full peer review and any attached files.

Reviewer #1: No

Reviewer #2: No

---

## [Author Response · Author response to Decision Letter 0]

24 Mar 2021

To the handling editor

Thank you for the helpful reviews. We have followed all suggestions, including up-dated figure 5. Please find attached the revised version.

Kind regards

Claus Rasmussen

---

## [Editor Report · Decision Letter 1]

31 Mar 2021

Evaluating competition for forage plants between honey bees and wild bees in Denmark

PONE-D-21-02226R1

Dear Dr. Rasmussen,

We’re pleased to inform you that your manuscript has been judged scientifically suitable for publication and will be formally accepted for publication once it meets all outstanding technical requirements.

Kind regards,

Fabio S. Nascimento

Academic Editor

PLOS ONE

---

## [Editor Report · Acceptance letter]

5 Apr 2021

PONE-D-21-02226R1 

Evaluating competition for forage plants between honey bees and wild bees in Denmark 

Dear Dr. Rasmussen:

I'm pleased to inform you that your manuscript has been deemed suitable for publication in PLOS ONE. Congratulations! Your manuscript is now with our production department. 

Kind regards, 

on behalf of

Dr. Fabio S. Nascimento 

Academic Editor

PLOS ONE